# SGD Learns Over-parameterized Networks that Provably Generalize on Linearly Separable Data

**Alon Brutzkus & Amir Globerson**
The Blavatnik School of Computer Science
Tel Aviv University, Israel
alonbrutzkus@mail.tau.ac.il,amir.globerson@gmail.com

**Eran Malach & Shai Shalev-Shwartz**
School of Computer Science
The Hebrew University, Israel
eran.malach@mail.huji.ac.il,shais@cs.huji.ac.il

## ABSTRACT

Neural networks exhibit good generalization behavior in the *over-parameterized* regime, where the number of network parameters exceeds the number of observations. Nonetheless, current generalization bounds for neural networks fail to explain this phenomenon. In an attempt to bridge this gap, we study the problem of learning a two-layer over-parameterized neural network, when the data is generated by a linearly separable function. In the case where the network has Leaky ReLU activations and only the first layer is trained, we provide both optimization and generalization guarantees for over-parameterized networks. Specifically, we prove convergence rates of SGD to a global minimum, and provide generalization guarantees for this global minimum that are independent of the network size. Therefore, our result clearly shows that the use of SGD for optimization both finds a global minimum, and avoids overfitting despite the high capacity of the model. This is the first theoretical demonstration that SGD can avoid overfitting, when learning over-specified neural network classifiers.

## 1 INTRODUCTION

Neural networks have achieved remarkable performance in many machine learning tasks. Although recently there have been numerous theoretical contributions to understand their success, it is still largely unexplained and remains a mystery. In particular, it is not known why in the over-parameterized setting, in which there are far more parameters than training points, stochastic gradient descent (SGD) can learn networks that generalize well, as been observed in practice (Neyshabur et al., 2014; Zhang et al., 2016).

In such over-parameterized settings, the loss function can contain multiple global minima that generalize poorly. Therefore, learning can in principle lead to models with low training error, but high test error. However, as often observed in practice, SGD is in fact able to find models with low training error *and* good generalization performance. This suggests that the optimization procedure, which depends on the optimization method (SGD) and the training data, introduces some form of *inductive bias* which directs it towards a low complexity solution. Thus, in order to explain the success of neural networks, it is crucial to characterize this inductive bias and understand what are the guarantees for generalization of over-parameterized neural networks.

In this work, we address these problems in a binary classification setting where SGD optimizes a two-layer over-parameterized network with the goal of learning a *linearly separable* function. We study a relatively simple case of SGD where the weights of the second layer are fixed throughout the training process, and only the weights of the first layer are updated. Clearly, an over-parameterized network is not necessary for classifying linearly separable data, since this is possible with linear

classifiers (e.g., with the Perceptron algorithm) which also have good generalization guarantees (Shalev-Shwartz & Ben-David, 2014). But, the key question which we address here is whether a large network will overfit in such a case or not. As we shall see, it turns out that although the networks we consider are rich enough to considerably overfit the data, this does not happen when SGD is used for optimization. In other words, SGD introduces an inductive bias which allows it to learn over-parameterized networks that can generalize well. Therefore, this setting serves as a good test bed for studying the effect of over-paramaterization.

## 2 PROBLEM FORMULATION

Define $\mathcal{X} = \{\boldsymbol{x} \in \mathbb{R}^d : \|\boldsymbol{x}\| \leq 1\}$, $\mathcal{Y} = \{\pm 1\}$. We consider a distribution over linearly separable points. Formally, let $\mathcal{D}$ be a distribution over $\mathcal{X} \times \mathcal{Y}$ such that there exists $\boldsymbol{w}^* \in \mathbb{R}^d$ for which $\mathbb{P}_{(\boldsymbol{x},y) \sim \mathcal{D}}(y \langle \boldsymbol{w}^*, \boldsymbol{x} \rangle \geq 1) = 1$. [1] Let $S = \{(\boldsymbol{x}_1, y_1), \ldots, (\boldsymbol{x}_n, y_n)\} \subseteq \mathcal{X} \times \mathcal{Y}$ be a training set sampled i.i.d. from $\mathcal{D}$. [2]

Consider the following two-layer neural network, with $2k > 0$ hidden units. [3] The network parameters are $W \in \mathbb{R}^{2k \times d}$, $\boldsymbol{v} \in \mathbb{R}^{2k}$, which we denote jointly by $\mathcal{W} = (W, \boldsymbol{v})$. The network output is given by the function $N_{\mathcal{W}} : \mathbb{R}^d \to \mathbb{R}$ defined as: [4]

$$N_{\mathcal{W}}(\boldsymbol{x}) = \boldsymbol{v}^\top \sigma(W\boldsymbol{x}) \tag{1}$$

where $\sigma$ is a non-linear activation function applied element-wise.

We define the empirical loss over $S$ to be the mean hinge-loss:

$$L_S(W) = \frac{1}{n} \sum_{i=1}^{n} \max\{1 - y_i N_{\mathcal{W}}(\boldsymbol{x}_i), 0\}$$

Note that for convenience of analysis, we will sometimes refer to $L_S$ as a function over a vector. Namely, for a matrix $W \in \mathbb{R}^{2k \times d}$, we will consider instead its vectorized version $\vec{W} \in \mathbb{R}^{2kd}$ (where the rows of $W$ are concatenated) and define, with abuse of notation, that $L_S(\vec{W}) = L_S(W)$.

In our setting we fix the second layer to be $\boldsymbol{v} = (\overbrace{v \ldots v}^{k}, \overbrace{-v \cdots - v}^{k})$ such that $v > 0$ and only learn the weight matrix $W$. We will consider only positive homogeneous activations (Leaky ReLU and ReLU) and thus the network we consider with $2k$ hidden neurons is as expressive as networks with $k$ hidden neurons and any vector $\boldsymbol{v}$ in the second layer. [5] Hence, we can fix the second layer without limiting the expressive power of the two-layer network. Although it is relatively simpler than the case where the second layer is not fixed, the effect of over-parameterization can be studied in this setting as well.

Hence, the objective of the optimization problem is to find:

$$\arg \min_{W \in \mathbb{R}^{2k \times d}} L_S(W) \tag{2}$$

where $\min_{W \in \mathbb{R}^{2k \times d}} L_S(W) = 0$ holds for the activations we will consider (Leaky ReLU and ReLU).

---

[1] This implies that $\|\boldsymbol{w}^*\| \geq 1$.

[2] Without loss of generality, we will ignore the event that $y_i \langle \boldsymbol{w}^*, \boldsymbol{x}_i \rangle < 1$ for some $i$, since this is an event of measure zero.

[3] We have an even number of hidden neurons for ease of exposition. See the definition of $\boldsymbol{v}$ below.

[4] Our results hold in the case where the first layer contains bias terms. This follows by the standard argument of adding another dimension to the input and setting the value 1 in the extra dimension for each data point.

[5] For example, consider a network with $k$ hidden neurons with positive homogeneous activations, where each hidden neuron $i$ has incoming weight vector $\boldsymbol{w}_i$ and outgoing weight $v_i$. Then, we can express this network with the network defined in Eq. 1 as follows. For each $i$ such that $v_i > 0$, we define a neuron in the new network with incoming weight vector $v_i \boldsymbol{w}_i$ and outgoing weight 1. Similarly, if $v_i < 0$, we define a neuron in the new network with incoming weight vector $-v_i \boldsymbol{w}_i$ and outgoing weight $-1$. For all other neurons in the new network we define an incoming zero weight vector. Due to the positive homogeneity, it follows that this network is equivalent to the network with $k$ hidden neurons.

We focus on the case where $L_S(W)$ is minimized using an SGD algorithm with batch of size 1, and where only the weights of the first layer (namely $W$) are updated. At iteration $t$, SGD randomly chooses a point $(\boldsymbol{x}_t, y_t) \in S$ and updates the weights with a constant learning rate $\eta$. Formally, let $\mathcal{W}_t = (W_t, \boldsymbol{v})$ be the parameters at iteration $t$, then the update at iteration $t$ is given by

$$W_t = W_{t-1} - \eta \frac{\partial}{\partial W} L_{\{(\boldsymbol{x}_t, y_t)\}}(W_{t-1}) \tag{3}$$

We define a *non-zero* update at iteration $t$ if it holds that $\frac{\partial}{\partial W} L_{\{(\boldsymbol{x}_t, y_t)\}}(W_{t-1}) \neq 0$. Finally, we will need the following notation. For $1 \leq i \leq k$, we denote by $\boldsymbol{w}_t^{(i)} \in \mathbb{R}^d$ the incoming weight vector of neuron $i$ at iteration $t$. [6] Similarly, for $1 \leq i \leq k$ we define $\boldsymbol{u}_t^{(i)} \in \mathbb{R}^d$ to be the incoming weight vector of neuron $k + i$ at iteration $t$.

## 3 MAIN RESULT

We now present our main results, for the case where $\sigma$ is the Leaky ReLU function. Namely, $\sigma(z) = \max\{\alpha z, z\}$ where $0 < \alpha < 1$.

First, we show that SGD can find a global optimum of $L_S(W)$. Note that this is by no means obvious, since $L_S(W)$ is a non-convex function (see Proposition 1). Specifically, we show that SGD converges to such an optimum while making at most:

$$M = \frac{\|\boldsymbol{w}^*\|^2}{\alpha^2} + O\left(\frac{\|\boldsymbol{w}^*\|^2}{\min\{\eta, \sqrt{\eta}\}}\right) \tag{4}$$

*non-zero* update steps (see Corollary 3). In particular, the bound is *independent* of the number of neurons $2k$. To the best of our knowledge, this is the first convergence guarantee of SGD for neural networks with the hinge loss. Furthermore, we prove a lower bound of $\Omega\left(\frac{\|\boldsymbol{w}^*\|}{\eta} + \|\boldsymbol{w}^*\|^2\right)$ for the number of non-zero updates (see Theorem 4).

Next, we address the question of generalization. As noted earlier, since the network is large, it can in principle overfit. Indeed, there are parameter settings for which the network will have arbitrarily bad test error (see Section 6.2). However, as we show here, this will not happen in our setting where SGD is used for optimization. In Theorem 6 we use a compression bound to show that the model learned by SGD will have a generalization error of $O\left(\frac{M \log n}{n}\right)$.[7] This implies that for *any* network size, given a sufficiently large number of training samples that is *independent* of the network size, SGD converges to a global minimum with good generalization behaviour. This is despite the fact that for sufficiently large $k$ there are multiple global minima which overfit the training set (see Section 6.2). This implies that SGD is biased towards solutions that can be expressed by a small set of training points and thus generalizes well.

To summarize, when the activation is the Leaky ReLU and the data is linearly separable, we provide provable guarantees of optimization, generalization and expressive power for over-parameterized networks. This allows us to provide a rigorous explanation of the performance of over-parameterized networks in this setting. This is a first step in unraveling the mystery of the success of over-parameterized networks in practice.

We further study the same over-parameterized setting where the non-linear activation is the ReLU function (i.e., $\sigma(z) = \max\{0, z\}$). Surprisingly, this case has different properties. Indeed, we show that the loss contains spurious local minima and thus the previous convergence result of SGD to a global minimum does not hold in this case. Furthermore, we show an example where over-parameterization is favorable from an optimization point of view. Namely, for a sufficiently small number of hidden neurons, SGD will converge to a *local* minimum with high probability, whereas for a sufficiently large number of hidden neurons, SGD will converge to a *global* minimum with high probability.

---

[6]These are the neurons with positive outgoing weight $v > 0$.

[7]See discussion in Remark 1 on the dependence of the generalizaion bound on $\eta$.

The paper is organized as follows. We discuss related work in Section 4 . In Section 5 we prove the convergence bounds, in Section 6 we give the generalization guarantees and in Section 7 the results for the ReLU activation. We conclude our work in Section 8.

## 4 RELATED WORK

The generalization performance of neural networks has been studied extensively. Earlier results (Anthony & Bartlett, 2009) provided bounds that depend on the VC dimension of the network, and the VC dimension was shown to scale linearly with the number of parameters. More recent works, study alternative notions of complexity, such as Rademacher compexity (Bartlett & Mendelson, 2002; Neyshabur et al., 2015; Bartlett et al., 2017; Kawaguchi et al., 2017), Robustness (Xu & Mannor, 2012) and PAC-Bayes (Neyshabur et al., 2017b). However, all of these notions do not provide *provable* guarantees for the generalization performance of over-parameterized networks trained with gradient based methods (Neyshabur et al., 2017a). The main disadvantage of these approaches, is that they do not depend on the optimization method (e.g., SGD), and thus do not capture its role in the generalization performance. In a recent paper, Dziugaite & Roy (2017) numerically optimize a PAC-Bayes bound of a stochastic over-parameterized network in a binary classification task and obtain a nonvacuous generalization bound. However, their bound is effective only when optimization succeeds, which their results do not guarantee. In our work, we give generalization guarantees based on a compression bound that follows from convergence rate guarantees of SGD, and thus take into account the effect of the optimization method on the generalization performance. This analysis results in generalization bounds that are independent of the network size and thus hold for over-parameterized networks.

Stability bounds for SGD in non-convex settings were given in Hardt et al. (2016); Kuzborskij & Lampert (2017). However, their results hold for smooth loss functions, whereas the loss function we consider is not smooth due to the non-smooth activation functions (Leaky ReLU, ReLU).

Other works have studied generalization of neural networks in a model recovery setting, where assumptions are made on the underlying model and the input distribution (Brutzkus & Globerson, 2017; Zhong et al., 2017; Li & Yuan, 2017; Du et al., 2017; Tian, 2017). However, in their works the neural networks are not over-parameterized as in our setting.

Soltanolkotabi et al. (2017) analyze the optimization landscape of over-parameterized networks and give convergence guarantees for gradient descent to a global minimum when the data follows a Gaussian distribution and the activation functions are differentiable. The main difference from our work is that they do not provide generalization guarantees for the resulting model. Furthermore, we do not make any assumptions on the distribution of the feature vectors.

In a recent work, Nguyen & Hein (2017) show that if training points are linearly separable then under assumptions on the rank of the weight matrices of a fully-connected neural network, every critical point of the loss function is a global minimum. Their work extends previous results in Gori & Tesi (1992); Frasconi et al. (1997); Yu & Chen (1995). Our work differs from these in several respects. First, we show global convergence guarantees of SGD, whereas they only analyze the optimization landscape, without direct implications on performance of optimization methods. Second, we provide generalization bounds and their focus is solely on optimization. Third, we consider non-differentiable activation functions (Leaky ReLU, ReLU) while their results hold only for continuously differentiable activation functions.

## 5 CONVERGENCE ANALYSIS

In this section we consider the setting of Section 2 with a leaky ReLU activation function. In Section 5.1 we show SGD will converge to a globally optimal solution, and analyze the rate of convergence. In Section 5.1 we also provide lower bounds on the rate of convergence. The results in this section are interesting for two reasons. First, they show convergence of SGD for a non-convex objective. Second, the rate of convergence results will be used to derive generalization bounds in Section 6.

## 5.1 Upper Bound

Before proving convergence of SGD to a global minimum, we show that every critical point is a global minimum and the loss function is non-convex. The proof is deferred to the appendix.

**Proposition 1.** $L_S(W)$ *satisfies the following properties: 1) Every critical point is a global minimum. 2) It is non-convex.*

Let $\vec{W}_t = (\boldsymbol{w}_t^{(1)} \ldots \boldsymbol{w}_t^{(k)} \boldsymbol{u}_t^{(1)} \ldots \boldsymbol{u}_t^{(k)}) \in \mathbb{R}^{2kd}$ be the vectorized version of $W_t$ and $N_t := N_{\mathcal{W}_t}$ where $\mathcal{W}_t = (W_t, \boldsymbol{v})$ (see Eq. 1). Since we will show an upper bound on the number of non-zero updates, we will assume for simplicity that for all $t$ we have a non-zero update at iteration $t$.

We assume that SGD is initialized such that the norms of all rows of $W_0$ are upper bounded by some constant $R > 0$. Namely for all $1 \le i \le k$ it holds that:

$$\|\boldsymbol{w}_0^{(i)}\|, \|\boldsymbol{u}_0^{(i)}\| \le R \tag{5}$$

Define $M_k := \frac{\|\boldsymbol{w}^*\|^2}{\alpha^2} + \frac{\|\boldsymbol{w}^*\|^2}{k\eta v^2 \alpha^2} + \frac{\sqrt{R(8k^2\eta^2 v^2 + 8\eta k)}\|\boldsymbol{w}^*\|^{1.5}}{2k(\eta v \alpha)^{1.5}} + \frac{2R\|\boldsymbol{w}^*\|}{\eta v \alpha}$. We give an upper bound on the number of non-zero updates SGD makes until convergence to a critical point (which is a global minimum by Proposition 1). The result is summarized in the following theorem.

**Theorem 2.** *SGD converges to a global minimum after performing at most $M_k$ non-zero updates.*

We will briefly sketch the proof of Theorem 2. The full proof is deferred to the Appendix (see Section 9.1.2). The analysis is reminiscent of the Perceptron convergence proof (e.g. in Shalev-Shwartz & Ben-David (2014)), but with key modifications due to the non-linear architecture. Concretely, assume SGD performed $t$ non-zero updates. We consider the vector $\vec{W}_t$ and the vector $\vec{W}^* = (\overbrace{\boldsymbol{w}^* \ldots \boldsymbol{w}^*}^{k}, \overbrace{-\boldsymbol{w}^* \cdots -\boldsymbol{w}^*}^{k}) \in \mathbb{R}^{2kd}$ which is a global minimum of $L_S$. We define $F(W_t) = \left\langle \vec{W}_t, \vec{W}^* \right\rangle$ and $G(W_t) = \|\vec{W}_t\|$. Then, we give an upper bound on $G(W_t)$ in terms of $G(W_{t-1})$ and by a recursive application of inequalities we show that $G(W_t)$ is bounded from above by a square root of a linear function of $t$. Similarly, by a recursive application of inequalities, we show that $F(W_t)$ is bounded from below by a linear function of $t$. Finally, we use the Cauchy-Schwartz inequality, $\frac{|F(W_t)|}{G(W_t)\|\vec{W}^*\|} \le 1$, to show that $t \le M_k$.

To obtain a simpler bound than the one obtained in Theorem 2, we use the fact that we can set $R, v$ arbitrarily, and choose:[8]

$$R = v = \frac{1}{\sqrt{2k}}. \tag{6}$$

Then by Theorem 2 we get the following. The derivation is given in the Appendix (Section 9.1.3).

**Corollary 3.** *Let $R = v = \frac{1}{\sqrt{2k}}$, then SGD converges to a global minimum after performing at most*

$$M_k = \frac{\|\boldsymbol{w}^*\|^2}{\alpha^2} + O\left(\frac{\|\boldsymbol{w}^*\|^2}{\min\{\eta, \sqrt{\eta}\}}\right) \text{ non-zero updates.}$$

Thus the bound consists of two terms, the first which only depends on the margin (via $\|\boldsymbol{w}^*\|$) and the second which scales inversely with $\eta$. More importantly, the bound is independent of the network size.

## 5.2 Lower Bound

We use the same notations as in Section 5.1. The lower bound is given in the following theorem, which is proved in the Appendix (Section 9.1.4).

**Theorem 4.** *Assume SGD is initialized according to Eq. 6, then for any $d$ there exists a sequence of linearly separable points on which SGD will make at least $\Omega\left(\frac{\|\boldsymbol{w}^*\|}{\eta} + \|\boldsymbol{w}^*\|^2\right)$ mistakes.*

---

[8]This initialization resembles other initializations that are used in practice (Bengio, 2012; Glorot & Bengio, 2010)

Although this lower bound is not tight, it does show that the upper bound in Corollary 3 cannot be much improved. Furthermore, the example presented in the proof of Theorem 4, demonstrates that $\eta \to \infty$ can be optimal in terms of optimization and generalization, i.e., SGD makes the minimum number of updates ($\|\boldsymbol{w}^*\|^2$) and the learned model is equivalent to the true classifier $\boldsymbol{w}^*$. We will use this observation in the discussion on the dependence of the generalization bound in Theorem 6 on $\eta$ (see Remark 1).

### 5.3 EXTENSIONS - UPDATING BOTH LAYERS

The bounds we provide in this section rely on the assumption that the weights of the second layer remain constant throughout the training process. Although this does not limit the expressive power of the network, updating both layers effectively changes the dynamics of the problem, and it may not be clear why the above bounds apply to this case as well. To answer this concern we show the following. First, we run the same experiments as in Figure 1, but with both layers trained. We show in Figure 2 that the training and generalization performance remain the same. Second, in the complete proof of the upper bound given in Section 9.1.2, we relax the assumption that the weights of the second layer are fixed, and only assume that they do not change signs during the training process, and that their absolute values are bounded from below and from above. This results in a similar bound, up to a constant factor. We corroborate our theoretical result with experiments and show in Figure 3 that by choosing an appropriate constant learning rate, this in fact holds when updating both layers - the weights of the last layer do not change their sign, and are correctly bounded. Furthermore, the performance of SGD is not affected by the choice of the learning rate. A complete theoretical analysis of training both layers is left for future work.

## 6 GENERALIZATION

In this section we give generalization guarantees for SGD learning of over-parameterized networks with Leaky ReLU activations. These results are obtained by combining Theorem 2 with a compression generalization bound (see Section 6.1). In Section 6.2 we show that over-parameterized networks are sufficiently expressive to contain global minima that overfit the training set. Taken together, these results show that although there are models that overfit, SGD effectively avoids these, and finds the models that generalize well.

### 6.1 COMPRESSION BOUND

Given the bound in Theorem 2 we can invoke compression bounds for generalization guarantees with respect to the 0-1 loss (Littlestone & Warmuth, 1986) . Denote by $N_k$ a two-layer neural network with $2k$ hidden neurons defined in Section 1 where $\sigma$ is the Leaky ReLU. Let $SGD_k(S, W_0)$ be the output of running SGD for training this network on a set $S$ and initialized with $W_0$ that satisfies Eq. 5. Define $\mathcal{H}_k$ to be the set of all possible hypotheses that $SGD_k(S, W_0)$ can output for *any* $S$ and $W_0$ which satisfies Eq. 5.

Now, fix an initialization $W_0$. Then the key observation is that by Theorem 2 we have $SGD_k(S, W_0) = B_{W_0}(\boldsymbol{x}_{i_1}, ..., \boldsymbol{x}_{i_{c_k}})$ for $c_k \leq M_k$, some function $B_{W_0} : \mathcal{X}^{c_k} \to \mathcal{H}_k$ and $(i_1, ..., i_{c_k}) \in [n]^{c_k}$.[9] Equivalently, $SGD_k(\cdot, W_0)$ and $B_{W_0}$ define a compression scheme of size $c_k$ for hypothesis class $\mathcal{H}_k$ (see Definition 30.4 in Shalev-Shwartz & Ben-David (2014)). Denote by $V = \{\boldsymbol{x}_j : j \notin \{i_1, ..., i_{c_k}\}\}$ the set of examples which were not selected to define $SGD_k(S, W_0)$. Let $L_{\mathcal{D}}^{0-1}(SGD_k(S, W_0))$ and $L_V^{0-1}(SGD_k(S, W_0))$ be the true risk of $SGD_k(S, W_0)$ and empirical risk of $SGD_k(S, W_0)$ on the set $V$, respectively. Then by Theorem 30.2 and Corollary 30.3 in Shalev-Shwartz & Ben-David (2014) we can easily derive the following theorem. The proof is deferred to the Appendix (Section 9.2.1).

**Theorem 5.** *Let $n \geq 2c_k$, then with probability of at least $1 - \delta$ over the choice of $S$ and $W_0$ we have*

$$L_{\mathcal{D}}^{0-1}(SGD_k(S, W_0)) \leq L_V^{0-1}(SGD_k(S, W_0)) + \sqrt{L_V^{0-1}(SGD_k(S, W_0))\frac{4c_k \log \frac{n}{\delta}}{n}} + \frac{8c_k \log \frac{n}{\delta}}{n}$$

---

[9]We use a subscript $W_0$ because the function is determined by $W_0$.

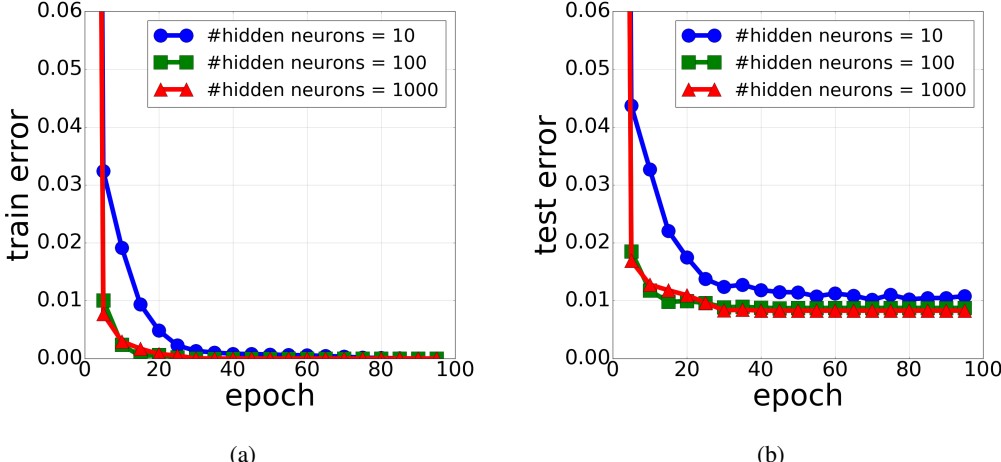

(a)                                        (b)

Figure 1: Classifying MNIST images with over-parameterized networks. The setting of Section 5 is implemented (e.g., SGD with batch of size 1, only first layer is trained, Leaky ReLU activations) and SGD is initialized according to the initialization defined in Eq. 6. The linearly separable data set consists of 4000 MNIST images with digits 3 and 5, each of dimension 784. The size of the training set is 3000 and the remaining 1000 points form the test set. Three experiments are performed which differ only in the number of hidden neurons, 10, 100 and 1000. In the latter two, the networks are over-parameterized. For each number of hidden neurons, 40 different runs of SGD are performed and their results are averaged. (a) shows that in all experiments SGD converges to a global minimum. (b) shows that the global minimum obtained by SGD generalizes well in all settings (including the over-parameterized).

Since $L_V^{0-1}(SGD_k(S, W_0)) = 0$ holds at a global minimum of $L_S$, then by Combining the results of Corollary 3 and Theorem 5, we get the following theorem.

**Theorem 6.** *If $n \geq 2c_k$ and assuming the initialization defined in Eq. 6, then with probability at least $1 - \delta$ over the choice of $S$ and $W_0$, SGD converges to a global minimum of $L_S$ with 0-1 test error at most*

$$\frac{8}{n}\left(\frac{\|\boldsymbol{w}^*\|^2}{\alpha^2} + O\left(\frac{\|\boldsymbol{w}^*\|^2}{\min\{\eta, \sqrt{\eta}\}}\right)\right)\log\frac{n}{\delta} \tag{7}$$

Thus for fixed $\|\boldsymbol{w}^*\|$ and $\eta$ we obtain a sample complexity guarantee that is independent of the network size (See Remark 1 for a discussion on the dependence of the bound on $\eta$). This is despite the fact that for sufficiently large $k$, the network has *global* minima that have arbitrarily high test errors, as we show in the next section. Thus, SGD and the linearly separable data introduce an inductive bias which directs SGD to the global minimum with low test error while avoiding global minima with high test error. In Figure 1 we demonstrate this empirically for a linearly separable data set (from a subset of MNIST) learned using over-parameterized networks. The figure indeed shows that SGD converges to a global minimum which generalizes well.

**Remark 1.** *The generelization bound in Eq. 7 holds for $\eta \to \infty$, which is unique for the setting that we consider, and may seem surprising, given that a choice of large $\eta$ often fails in practice. Furthermore, the bound is optimal for $\eta \to \infty$. To support this theoretical result, we show in Theorem 4 an example where indeed $\eta \to \infty$ is optimal in terms of the number of updates and generalization. On the other hand, we note that in practice, it may not be optimal to use large $\eta$ in our setting, since this bound results from a worst-case analysis of a sequence of examples encountered by SGD. Finally, the important thing to note is that the bound holds for any $\eta$, and is thus applicable to realistic applications of SGD.*

## 6.2 Expressiveness

Let $X \in \mathbb{R}^{d \times n}$ be the matrix with the points $\boldsymbol{x}_i$ in its columns, $\boldsymbol{y} \in \{-1, 1\}^n$ the corresponding vector of labels and let $N_{\mathcal{W}}(X) = \boldsymbol{v}^\top \sigma(WX)$ be the network defined in Eq. 1 applied on the matrix $X$. By Theorem 8 in (Soudry & Hoffer, 2017) we immediately get the following. For completeness, the proof is given in the Appendix (Section 9.2.2).

**Theorem 7.** *Assume that $k \geq 2 \left\lceil \frac{n}{2d-2} \right\rceil$. Then for any $\boldsymbol{y} \in \{-1, 1\}^n$ and for almost any $X$,[10]*

*there exist $\tilde{\mathcal{W}} = (\tilde{W}, \tilde{\boldsymbol{v}})$ where $\tilde{W} \in \mathbb{R}^{2k \times d}$ and $\tilde{\boldsymbol{v}} = (\overbrace{\tilde{v} \ldots \tilde{v}}^{k}, \overbrace{-\tilde{v} \cdots -\tilde{v}}^{k}) \in \mathbb{R}^{2k}$, $\tilde{v} > 0$ such that $\boldsymbol{y} = N_{\tilde{\mathcal{W}}}(X)$.*

Theorem 7 implies that for sufficiently large networks, the optimization problem (2) can have arbitrarely *bad* global minima with respect to a given test set, i.e., ones which do not generalize well on a given test set.

## 7 ReLU- Success and Failure Cases

In this section we consider the same setting as in section 5, but with the ReLU activation function $\sigma(x) = \max\{0, x\}$. In Section 7.1 we show that the loss function contains arbitrarely bad local minima. In Section 7.2 we give an example where for a sufficiently small network, with high probability SGD will converge to a local minimum. On the other hand, for a sufficiently large network, with high probability SGD will converge to a global minimum.

### 7.1 Existence of Bad Local Minima

The result is summarized in the following theorem and the proof is deferred to the Appendix (Section 9.3.1). The main idea is to construct a network with weight paramater $W$ such that for at least $\frac{|S|}{2}$ points $(\boldsymbol{x}, y) \in S$ it holds that $\langle \boldsymbol{w}, \boldsymbol{x} \rangle < 0$ for each neuron with weight vector $\boldsymbol{w}$. Furthermore, the remaining points satisfy $y N_{\mathcal{W}}(\boldsymbol{x}) > 1$ and thus the gradient is zero and $L_S(W) > \frac{1}{2}$.

**Theorem 8.** *Fix $\boldsymbol{v} = (\overbrace{1 \ldots 1}^{k}, \overbrace{-1 \cdots -1}^{k}) \in \mathbb{R}^{2k}$. Then, for every finite set of examples $S \subseteq \mathcal{X} \times \mathcal{Y}$ that is linearly separable, i.e., for which there exists $\boldsymbol{w}^* \in \mathbb{R}^d$ such that for each $(\boldsymbol{x}, y) \in S$ we have $y \langle \boldsymbol{w}^*, \boldsymbol{x} \rangle \geq 1$, there exists $W \in \mathbb{R}^{2k \times d}$ such that $W$ is a local minimum point with $L_S(W) > \frac{1}{2}$.*

### 7.2 Orthogonal Vectors - Simple Case Analysis

In this section we assume that $S = \{\boldsymbol{e}_1 \ldots \boldsymbol{e}_d\} \times \{1\} \subseteq \mathcal{X} \times \mathcal{Y}$ where $\{\boldsymbol{e}_1, \ldots, \boldsymbol{e}_d\}$ is the standard basis of $\mathbb{R}^d$. We assume all examples are labeled with the same label for simplicity, as the same result holds for the general case.

Let $N_{\mathcal{W}_t}$ be the network obtained at iteration $t$, where $\mathcal{W}_t = (W_t, \boldsymbol{v})$. Assume we initialize with fixed $\boldsymbol{v} = (\overbrace{1 \ldots 1}^{k}, \overbrace{-1 \cdots -1}^{k})$, and $W_0 \in \mathbb{R}^{2k \times d}$ is randomly initialized from a continuous symmetric distribution with bounded norm, i.e $|[W_0]_{i,j}| \leq C$ for some $C > 0$.

The main result of this section is given in the following theorem. The proof is given in the Appendix (Section 9.3.2). The main observation is that the convergence to non-global minimum depends solely on the initialization and occurs if and only if there exists a point $\boldsymbol{x}$ such that for all neurons, the corresponding initialized weight vector $\boldsymbol{w}$ satisfies $\langle \boldsymbol{w}, \boldsymbol{x} \rangle \leq 0$.

**Theorem 9.** *Fix $\delta > 0$ and assume we run SGD with examples from $S = \{\boldsymbol{e}_1 \ldots \boldsymbol{e}_d\} \times \{1\}$. If $k \leq \log_2(\frac{d}{-\ln(\delta)})$, then with probability of at least $1 - \delta$, SGD will converge to a non global minimum point.*
*On the other hand, if $k \geq \log_2(\frac{2d}{\delta})$, then with probability of at least $1 - \delta$, SGD will converge to a global minimum point after $\lceil \max\{\frac{dC}{\eta}, \frac{d}{\eta}\} \rceil$ iterations.*

---

[10]That is, the set of entries of $X$ which do not satisfy the statement is of Lebesgue measure 0.

Note that in the first part of the theorem, we can make the basin of attraction of the non-global minimum exponentially large by setting $\delta = e^{-\alpha d}$ for $\alpha \leq \frac{1}{2}$.

## 8 CONCLUSION

Understanding the performance of over-parameterized neural networks is essential for explaining the success of deep learning models in practice. Despite a plethora of theoretical results for generalization of neural networks, none of them give guarantees for over-parameterized networks. In this work, we give the first provable guarantees for the generalization performance of over-parameterized networks, in a setting where the data is linearly separable and the network has Leaky ReLU activations. We show that SGD compresses its output when learning over-parameterized networks, and thus exhibits good generalization performance.

The analysis for networks with Leaky ReLU activations does not hold for networks with ReLU activations, since in this case the loss contains spurious local minima. However, due to the success of over-parameterized networks with ReLU activations in practice, it is likely that similar results hold here as well. It would be very interesting to provide convergence guarantees and generalization bounds for this case. Another direction for future work is to show that similar results hold under different assumptions on the data.

### ACKNOWLEDGMENTS

This research is supported by the Blavatnik Computer Science Research Fund, ISF F.I.R.S.T. (Bikura) grant and the European Research Council (TheoryDL project).

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

# 9 APPENDIX

## 9.1 MISSING PROOFS FOR SECTION 5

### 9.1.1 PROOF OF PROPOSITION 1

1. Denote by $\vec{W} = \left( \boldsymbol{w}^{(1)} \ldots \boldsymbol{w}^{(k)} \boldsymbol{u}^{(1)} \ldots \boldsymbol{u}^{(k)} \right) \in \mathbb{R}^{2kd}$ the vector of all parameters where each $\boldsymbol{w}^{(i)}, \boldsymbol{u}^{(i)} \in \mathbb{R}^d$. Let $(\boldsymbol{x}, y) \in S$, then if $y N_{\mathcal{W}}(\boldsymbol{x}) < 1$, it holds that

$$\left\langle \frac{\partial}{\partial \boldsymbol{w}^{(i)}} L_{\{(\boldsymbol{x},y)\}}(\vec{W}), \boldsymbol{w}^* \right\rangle = \left\langle -y\sigma' \left( \left\langle \boldsymbol{w}^{(i)}, \boldsymbol{x} \right\rangle \right) \boldsymbol{x}, \boldsymbol{w}^* \right\rangle \le -\sigma' \left( \left\langle \boldsymbol{w}^{(i)}, \boldsymbol{x} \right\rangle \right) < 0$$

and similarly,

$$\left\langle \frac{\partial}{\partial \boldsymbol{u}^{(i)}} L_{\{(\boldsymbol{x},y)\}}(\vec{W}), -\boldsymbol{w}^* \right\rangle = \left\langle y\sigma'\left(\left\langle \boldsymbol{u}^{(i)}, \boldsymbol{x}\right\rangle\right) \boldsymbol{x}, -\boldsymbol{w}^* \right\rangle \leq -\sigma'\left(\left\langle \boldsymbol{w}^{(i)}, \boldsymbol{x}\right\rangle\right) < 0.$$

Hence if we define $\vec{W}^* = (\overbrace{\boldsymbol{w}^* \ldots \boldsymbol{w}^*}^{k}, \overbrace{-\boldsymbol{w}^* \cdots -\boldsymbol{w}^*}^{k}) \in \mathbb{R}^{2kd}$, then

$$\left\langle \frac{\partial}{\partial \vec{W}} L_{\{(\boldsymbol{x},y)\}}(\vec{W}), \vec{W}^* \right\rangle < 0$$

Otherwise, if $yN_{\mathcal{W}}(\boldsymbol{x}) \geq 1$, then the gradient vanishes and thus

$$\left\langle \frac{\partial}{\partial \vec{W}} L_{\{(\boldsymbol{x},y)\}}(\vec{W}), \vec{W}^* \right\rangle = 0$$

It follows that if there exists $(\boldsymbol{x}, y) \in S$, such that $yN_{\mathcal{W}}(\boldsymbol{x}) < 1$, then we have

$$\left\langle \frac{\partial}{\partial \vec{W}} L_S(\vec{W}), \vec{W}^* \right\rangle = \frac{1}{n} \sum_{i=1}^{n} \left\langle \frac{\partial}{\partial \vec{W}} L_{\{(\boldsymbol{x}_i, y_i)\}}(\vec{W}), \vec{W}^* \right\rangle < 0$$

and thus $\frac{\partial}{\partial \vec{W}} L_S(\vec{W}) \neq 0$. Therefore, for any critical point it holds that $yN_{\mathcal{W}}(\boldsymbol{x}) \geq 1$ for all $(\boldsymbol{x}, y) \in S$, which implies that it is a global minimum.

2. For simplicity consider the function $f_{\boldsymbol{x}}(\boldsymbol{w}, \boldsymbol{u}) = \sigma(\langle \boldsymbol{w}, \boldsymbol{x} \rangle) - \sigma(\langle \boldsymbol{u}, \boldsymbol{x} \rangle)$ for $\boldsymbol{x} \neq 0$. Define $\boldsymbol{w}_1 = \boldsymbol{w}_2 = \boldsymbol{u}_1 = \boldsymbol{x}$ and $\boldsymbol{u}_2 = -\boldsymbol{x}$. Then

$$f_{\boldsymbol{x}}(\boldsymbol{w}_1, \boldsymbol{u}_1) = 0$$

$$f_{\boldsymbol{x}}(\boldsymbol{w}_2, \boldsymbol{u}_2) = (1 + \alpha)\|\boldsymbol{x}\|^2$$

and

$$f_{\boldsymbol{x}}(\frac{\boldsymbol{w}_1 + \boldsymbol{w}_2}{2}, \frac{\boldsymbol{u}_1 + \boldsymbol{u}_2}{2}) = \|\boldsymbol{x}\|^2$$

and thus $f_{\boldsymbol{x}}(\frac{\boldsymbol{w}_1 + \boldsymbol{w}_2}{2}, \frac{\boldsymbol{u}_1 + \boldsymbol{u}_2}{2}) > \frac{1}{2} f_{\boldsymbol{x}}(\boldsymbol{w}_1, \boldsymbol{u}_1) + \frac{1}{2} f_{\boldsymbol{x}}(\boldsymbol{w}_2, \boldsymbol{u}_2)$ which implies that the function is not convex.

### 9.1.2 PROOF OF THEOREM 2

We will start by analyzing a case with more relaxed assumptions - namely, we do not assume that the weights of the second layer are fixed, but rather that they do not change signs, and are bounded in absolute value. Formally, let $v_t^{(i)}$ be the weight of the second layer neuron corresponding to the weight vector $\boldsymbol{w}_t^{(i)}$, and $\hat{v}_t^{(i)}$ the weight corresponding to $\boldsymbol{u}_t^{(i)}$. Then we assume there exist $c, C > 0$ such that:

$$c \leq \frac{v_t^{(i)}}{v_0^{(i)}} \leq C, \ c \leq \frac{\hat{v}_t^{(i)}}{\hat{v}_0^{(i)}} \leq C \tag{8}$$

And note that we take $v_0^{(i)} = -\hat{v}_0^{(i)} = v$.

Assume SGD performed $t$ non-zero updates. We will show that $t \leq M_k$. We note that if there is no $(\boldsymbol{x}, y) \in S$ such that the corresponding update is non-zero, then SGD has reached a critical point of $L_S$ (which is a global minimum by Proposition 1). Let $\vec{W}^* = (\overbrace{\boldsymbol{w}^* \ldots \boldsymbol{w}^*}^{k}, \overbrace{-\boldsymbol{w}^* \cdots -\boldsymbol{w}^*}^{k}) \in \mathbb{R}^{2kd}$ and note that $L_S(\vec{W}^*) = 0$, i.e., $\vec{W}^*$ is a global minimum. Define the following two functions:

$$F(W_t) = \left\langle \vec{W}_t, \vec{W}^* \right\rangle = \sum_{i=1}^{k} \left\langle \boldsymbol{w}_t^{(i)}, \boldsymbol{w}^* \right\rangle - \sum_{i=1}^{k} \left\langle \boldsymbol{u}_t^{(i)}, \boldsymbol{w}^* \right\rangle$$

$$G(W_t) = \|\vec{W}_t\| = \sqrt{\sum_{i=1}^{k} \|\boldsymbol{w}_t^{(i)}\|^2 + \sum_{i=1}^{k} \|\boldsymbol{u}_t^{(i)}\|^2}$$

Then, from Cauchy-Schwartz inequality we have

$$\frac{|F(W_t)|}{G(W_t)\|\vec{W}^*\|} = \frac{\left|\left\langle \vec{W}_t, \vec{W}^* \right\rangle\right|}{\|\vec{W}_t\|\|\vec{W}^*\|} \leq 1 \tag{9}$$

Since the update at iteration $t$ is non-zero, we have $y_t N_{t-1}(\boldsymbol{x}_t) < 1$ and the update rule is given by

$$\boldsymbol{w}_t^{(i)} = \boldsymbol{w}_{t-1}^{(i)} + \eta v_t^{(i)} p_t^{(i)} y_t \boldsymbol{x}_t \ , \ \ \boldsymbol{u}_t^{(i)} = \boldsymbol{u}_{t-1}^{(i)} - \eta \hat{v}_t^{(i)} q_t^{(i)} y_t \boldsymbol{x}_t \tag{10}$$

where $p_t^{(i)} = 1$ if $\left\langle \boldsymbol{w}_{t-1}^{(i)}, \boldsymbol{x}_t \right\rangle \geq 0$ and $p_t^{(i)} = \alpha$ otherwise. Similarly $q_t^{(i)} = 1$ if $\left\langle \boldsymbol{u}_{t-1}^{(i)}, \boldsymbol{x}_t \right\rangle \geq 0$ and $q_t^{(i)} = \alpha$ otherwise. It follows that:

$$G(W_t)^2 = \sum_{i=1}^{k} \|\boldsymbol{w}_t^{(i)}\|^2 + \sum_{i=1}^{k} \|\boldsymbol{u}_t^{(i)}\|^2$$

$$\leq \sum_{i=1}^{k} \|\boldsymbol{w}_{t-1}^{(i)}\|^2 + \sum_{i=1}^{k} \|\boldsymbol{u}_{t-1}^{(i)}\|^2 + 2\eta y_t \left( \sum_{i=1}^{k} \left\langle \boldsymbol{w}_{t-1}^{(i)}, \boldsymbol{x}_t \right\rangle v_t^{(i)} p_t^{(i)} - \sum_{i=1}^{k} \left\langle \boldsymbol{u}_{t-1}^{(i)}, \boldsymbol{x}_t \right\rangle \hat{v}_t^{(i)} q_t^{(i)} \right) + 2k\eta^2 C^2 v^2 \|\boldsymbol{x}_t\|^2$$

$$< \sum_{i=1}^{k} \|\boldsymbol{w}_{t-1}^{(i)}\|^2 + \sum_{i=1}^{k} \|\boldsymbol{u}_{t-1}^{(i)}\|^2 + 2\eta + 2k\eta^2 v^2 = G(W_{t-1})^2 + 2\eta + 2k\eta^2 C^2 v^2$$

where the second inequality follows since $y_t \left( \sum_{i=1}^{k} \left\langle \boldsymbol{w}_{t-1}^{(i)}, \boldsymbol{x}_t \right\rangle v_t^{(i)} p_t^{(i)} - \sum_{i=1}^{k} \left\langle \boldsymbol{u}_{t-1}^{(i)}, \boldsymbol{x}_t \right\rangle \hat{v}_t^{(i)} q_t^{(i)} \right) = y_t N_{t-1}(\boldsymbol{x}_t) < 1$. Using the above recursively, we obtain:

$$G(W_t)^2 \leq G(W_0)^2 + t(2k\eta^2 C^2 v^2 + 2\eta) \tag{11}$$

On the other hand,

$$F(W_t) = \sum_{i=1}^{k} \left\langle \boldsymbol{w}_t^{(i)}, \boldsymbol{w}^* \right\rangle - \sum_{i=1}^{k} \left\langle \boldsymbol{u}_t^{(i)}, \boldsymbol{w}^* \right\rangle$$

$$= \sum_{i=1}^{k} \left\langle \boldsymbol{w}_{t-1}^{(i)}, \boldsymbol{w}^* \right\rangle - \sum_{i=1}^{k} \left\langle \boldsymbol{u}_{t-1}^{(i)}, \boldsymbol{w}^* \right\rangle + \eta \sum_{i=1}^{k} \langle y_t \boldsymbol{x}_t, \boldsymbol{w}^* \rangle v_t^{(i)} p_t^{(i)} + \eta \sum_{i=1}^{k} \langle y_t \boldsymbol{x}_t, \boldsymbol{w}^* \rangle \hat{v}_t^{(i)} q_t^{(i)}$$

$$\geq \sum_{i=1}^{k} \left\langle \boldsymbol{w}_{t-1}^{(i)}, \boldsymbol{w}^* \right\rangle - \sum_{i=1}^{k} \left\langle \boldsymbol{u}_{t-1}^{(i)}, \boldsymbol{w}^* \right\rangle + 2k\eta cv\alpha = F(W_{t-1}) + 2k\eta cv\alpha$$

where the inequality follows since $\langle y_t \boldsymbol{x}_t, \boldsymbol{w}^* \rangle \geq 1$. This implies that

$$F(W_t) \geq F(W_0) + 2k\eta cv\alpha t \tag{12}$$

By combining equations Eq. 9, Eq. 11 and Eq. 12 we get,

$$-G(W_0)\|\vec{W}^*\| + 2k\eta cv\alpha t \leq F(W_0) + 2k\eta cv\alpha t \leq F(W_t) \leq \|\vec{W}^*\|G(W_t)$$

$$\leq \|\vec{W}^*\|\sqrt{G(W_0)^2 + t(2k\eta^2 C^2 v^2 + 2\eta)}$$

Using $\sqrt{a+b} \leq \sqrt{a} + \sqrt{b}$ the above implies,

$$-G(W_0)\|\vec{W}^*\| + 2k\eta cv\alpha t \leq \|\vec{W}^*\|G(W_0) + \|\vec{W}^*\|\sqrt{t}\sqrt{2k\eta^2 C^2 v^2 + 2\eta}$$

Since $\|\boldsymbol{w}_0^{(i)}\|, \|\boldsymbol{u}_0^{(i)}\| \leq R$ we have $G(W_0) \leq \sqrt{2k}R$. Noting that $\|\vec{W}^*\| = \sqrt{2k}\|\boldsymbol{w}^*\|$ we get,

$$at \leq b\sqrt{t} + c$$

where $a = 2k\eta cv\alpha$, $b = \sqrt{(4k^2\eta^2 C^2 v^2 + 4\eta k)}\|\boldsymbol{w}^*\|$ and $c = 4kR\|\boldsymbol{w}^*\|$. By inspecting the roots of the parabola $P(x) = x^2 - \frac{b}{a}x - \frac{c}{a}$ we conclude that

$$t \leq \left(\frac{b}{a}\right)^2 + \sqrt{\frac{c}{a}}\frac{b}{a} + \frac{c}{a} = \frac{(4k^2\eta^2 C^2 v^2 + 4\eta k)\|\boldsymbol{w}^*\|^2}{4k^2\eta^2 c^2 v^2 \alpha^2} + \frac{\sqrt{(4k^2\eta^2 C^2 v^2 + 4\eta k)}\|\boldsymbol{w}^*\|}{2k\eta cv\alpha}\sqrt{\frac{2R\|\boldsymbol{w}^*\|}{\eta cv\alpha}} + \frac{2R\|\boldsymbol{w}^*\|}{\eta cv\alpha}$$

$$= \frac{C^2\|\boldsymbol{w}^*\|^2}{c^2\alpha^2} + \frac{\|\boldsymbol{w}^*\|^2}{k\eta c^2 v^2 \alpha^2} + \frac{\sqrt{R(8k^2\eta^2 C^2 v^2 + 8\eta k)}\|\boldsymbol{w}^*\|^{1.5}}{2k(\eta cv\alpha)^{1.5}} + \frac{2R\|\boldsymbol{w}^*\|}{\eta cv\alpha} \tag{13}$$

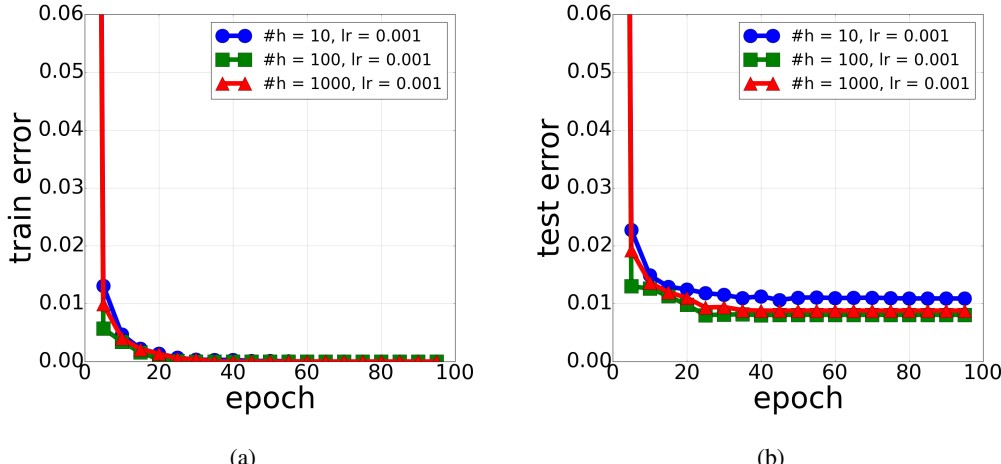

(a)                                                                      (b)

Figure 2: Classifying MNIST images with over-parameterized networks and training both layers. The setting of Figure 1 is implemented, but now the second layer is trained as well. The second layer is initialized as in Figure 1, i.e., all the weights are initialized to $\frac{1}{\sqrt{2k}}$ or $-\frac{1}{\sqrt{2k}}$. The training and generalization performance are similar to the performance in the case where only the first layer is trained (see Figure 1).

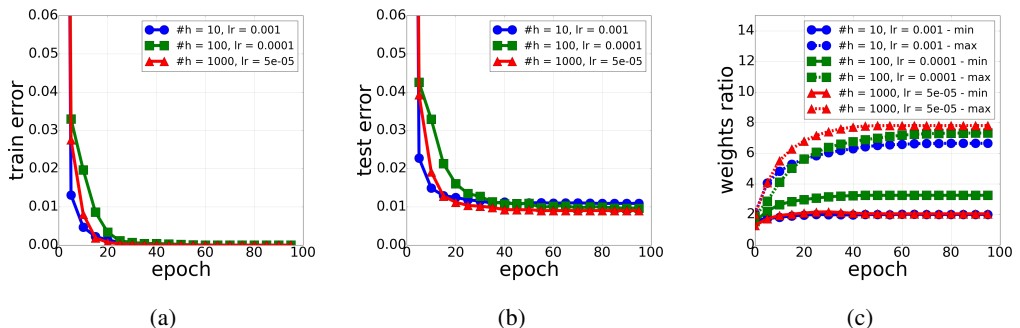

(a)                                         (b)                                         (c)

Figure 3: Classifying MNIST images with over-parameterized networks, training both layers and choosing an appropriate learning rate. The setting of Figure 2 is implemented, but here a different learning rate is chosen for each network size, in order to satisfy the conditions of the proof in Section 9.1.2. Figures (a) and (b) are train and test errors of MNIST classification for different network sizes and the chosen learning rates. In this setting, SGD exhibits similar training and generalization performance as in Figure 2. Figure (c) shows the minimal and maximal value of the second layer weights divided by their initial value (denoted as $c, C$ respectively in Section 9.1.2). It can be seen that these values remain above zero, which implies that the weights do not flip signs during the training process (namely they satisfy the sign condition in Section 9.1.2) and that they behave similarly for different network sizes.

Notice that when assuming that the weights of the second layer are fixed, we get $c = C = 1$ and the above is simply equal to $M_k$. Otherwise, if $c, C$ are independent constants, we get a similar bound, up to a constant factor.

### 9.1.3 PROOF OF COROLLARY 3

Since $\frac{R}{v} = 1$, we have by Theorem 2 and the inequality $\sqrt{a+b} \leq \sqrt{a} + \sqrt{b}$,

$$M_k = \frac{\|\boldsymbol{w}^*\|^2}{\alpha^2} + O\left(\frac{\|\boldsymbol{w}^*\|^2}{\eta}\right) + O\left(\frac{\|\boldsymbol{w}^*\|^{1.5}}{\sqrt{\eta}}\right) + O\left(\frac{\|\boldsymbol{w}^*\|^{1.5}}{\eta}\right) + O\left(\frac{\|\boldsymbol{w}^*\|}{\eta}\right)$$

$$= \frac{\|\boldsymbol{w}^*\|^2}{\alpha^2} + O\left(\frac{\|\boldsymbol{w}^*\|^2}{\min\{\eta, \sqrt{\eta}\}}\right). \tag{14}$$

### 9.1.4 PROOF OF THEOREM 4

We will prove a more general theorem. Theorem 4 follows by setting $R = v = \frac{1}{\sqrt{2k}}$.

**Theorem 10.** *For any $d$ there exists a sequence of linearly separable points on which SGD will make at least*

$$\max\left\{\min\left\{B_1, B_2\right\}, \|\boldsymbol{w}^*\|^2\right\}$$

*updates, where*

$$B_1 = \frac{R\|\boldsymbol{w}^*\|}{\eta v \alpha} + \min\left\{\frac{\|\boldsymbol{w}^*\|^2}{2\eta k v^2} - \alpha\|\boldsymbol{w}^*\|, 0\right\}$$

*and*

$$B_2 = \frac{R\|\boldsymbol{w}^*\|}{\eta v} + \min\left\{\frac{\|\boldsymbol{w}^*\|^2}{2\alpha^2 \eta k v^2} - \frac{\|\boldsymbol{w}^*\|}{\alpha}, 0\right\}$$

*Proof.* Define a sequence $\mathcal{S}$ of size $d$,

$$(\boldsymbol{e}_1, 1), (\boldsymbol{e}_2, 1), ..., (\boldsymbol{e}_d, 1)$$

where $\{\boldsymbol{e}_i\}$ is the standard basis of $\mathbb{R}^d$ and let $\boldsymbol{w}^* = (1, 1, ..., 1) \in \mathbb{R}^d$. Note that $d = \|\boldsymbol{w}^*\|^2$ and $\langle \boldsymbol{w}^*, \boldsymbol{e}_i \rangle \geq 1$ for all $1 \leq i \leq d$. We will consider the case where SGD runs on a sequence of examples which consists of multiple copies of $\mathcal{S}$ one after the other.

Assume SGD is initialized with

$$\boldsymbol{w}_0^{(i)} = -\sum_{j=1}^{d} \frac{R}{\sqrt{d}} \boldsymbol{e}_j$$

$$\boldsymbol{u}_0^{(i)} = \sum_{j=1}^{d} \frac{R}{\sqrt{d}} \boldsymbol{e}_j$$

for all $1 \leq i \leq k$. Note that $\|\boldsymbol{w}_0^{(i)}\|, \|\boldsymbol{u}_0^{(i)}\| \leq R$ for all $1 \leq i \leq k$.

Since $\boldsymbol{w}_0^{(i)} = \boldsymbol{w}_0^{(j)}$ and $\boldsymbol{u}_0^{(i)} = \boldsymbol{u}_0^{(j)}$ for all $i \neq j$, we have by induction that $\boldsymbol{w}_t^{(i)} = \boldsymbol{w}_t^{(j)}$ and $\boldsymbol{u}_t^{(i)} = \boldsymbol{u}_t^{(j)}$ for all $i \neq j$ and $t > 0$. Hence, we will denote $\boldsymbol{w}_t = \boldsymbol{w}_t^{(i)}$ and $\boldsymbol{u}_t = \boldsymbol{u}_t^{(i)}$ for all $1 \leq i \leq d$ and $t > 0$. Then for all $1 \leq j \leq d$ we have $\mathcal{N}_{\mathcal{W}_t}(\boldsymbol{e}_j) = kv\sigma'(\langle \boldsymbol{w}_t, \boldsymbol{e}_j \rangle) \langle \boldsymbol{w}_t, \boldsymbol{e}_j \rangle - kv\sigma'(\langle \boldsymbol{u}_t, \boldsymbol{e}_j \rangle) \langle \boldsymbol{u}_t, \boldsymbol{e}_j \rangle$.

At the global minumum $\mathcal{N}_{W_t, v}(\boldsymbol{e}_j) \geq 1$ for all $1 \leq j \leq d$, thus it follows that a necessary condition for convergence to a global minimum is that there exists an iteration $t$ in which either $kv\sigma'(\langle \boldsymbol{w}_t, \boldsymbol{e}_d \rangle) \langle \boldsymbol{w}_t, \boldsymbol{e}_d \rangle \geq \frac{1}{2}$ or $-kv\sigma'(\langle \boldsymbol{u}_t, \boldsymbol{e}_d \rangle) \langle \boldsymbol{u}_t, \boldsymbol{e}_d \rangle \geq \frac{1}{2}$. Equivalently, either $\langle \boldsymbol{w}_t, \boldsymbol{e}_d \rangle \geq \frac{1}{2kv}$ or $\langle \boldsymbol{u}_t, \boldsymbol{e}_d \rangle \leq -\frac{1}{2\alpha kv}$.

Since $\langle \boldsymbol{w}_0, \boldsymbol{e}_d \rangle = -\frac{R}{\sqrt{d}}$, then by the gradient updates (Eq. 10) it follows that after at least $\frac{R}{\eta v \alpha \sqrt{d}}$ copies of $\mathcal{S}$, or equivalently, after at least $\frac{Rd}{\eta v \alpha \sqrt{d}}$ iterations we will have $0 \leq \langle \boldsymbol{w}_t, \boldsymbol{e}_d \rangle \leq \eta v \alpha$. Then, after at least $\frac{d \min\{\frac{1}{2kv} - \eta v \alpha, 0\}}{\eta v}$ iterations we have $\langle \boldsymbol{w}_t, \boldsymbol{e}_d \rangle \geq \frac{1}{2kv}$. Thus, in total, after at least $\frac{R\|\boldsymbol{w}^*\|}{\eta v \alpha} + \min\{\frac{\|\boldsymbol{w}^*\|^2}{2\eta k v^2} - \alpha\|\boldsymbol{w}^*\|^2, 0\}$ iterations, we have $\langle \boldsymbol{w}_t, \boldsymbol{e}_d \rangle \geq \frac{1}{2kv}$.

By the same reasoning, we have $\langle \boldsymbol{u}_t, \boldsymbol{e}_d \rangle \leq -\frac{1}{2\alpha kv}$ after at least $\frac{R\|\boldsymbol{w}^*\|}{\eta v} + \min\{\frac{\|\boldsymbol{w}^*\|^2}{2\alpha^2 \eta k v^2} - \frac{\|\boldsymbol{w}^*\|}{\alpha}, 0\}$ iterations. Finally, SGD must update on at least $d$ points in order to converge to the global minimum. The claim now follows. $\qquad\square$

### 9.2 MISSING PROOFS FOR SECTION 6

#### 9.2.1 PROOF OF THEOREM 5

By Theorem 30.2 and Corollary 30.3 in Shalev-Shwartz & Ben-David (2014), for $n \geq 2c_k$ we have that with probability of at least $1 - \delta$ over the choice of $S$

$$L_{\mathcal{D}}(SGD_k(S, W_0)) \leq L_V(SGD_k(S, W_0)) + \sqrt{L_V(SGD_k(S, W_0)) \frac{4c_k \log \frac{n}{\delta}}{n}} + \frac{8c_k \log \frac{n}{\delta}}{n} \quad (15)$$

The above result holds for a *fixed* initialization $W_0$. We will show that the same result holds with high probability over $S$ *and* $W_0$, where $W_0$ is chosen independently of $S$ and satisfies Eq. 5. Define $\mathcal{B}$ to be the event that the inequality Eq. 15 does not hold. Then we know that $\mathbb{P}_S(\mathcal{B}|W_0) \leq \delta$ for any fixed initialization $W_0$. [11] Hence, by the law of total expectation,

$$\mathbb{P}_{S,W_0}(\mathcal{B}) = \mathbb{E}_{W_0}\left[\mathbb{P}_S(\mathcal{B}|W_0)\right] \leq \delta$$

#### 9.2.2 PROOF OF THEOREM 7

We can easily extend Theorem 8 in (Soudry & Hoffer, 2017) to hold for labels in $\{-1, 1\}$. By the theorem we can construct networks $N_{\mathcal{W}_1}$ and $N_{\mathcal{W}_2}$ such that for all $i$:

1. $N_{\mathcal{W}_1}(\boldsymbol{x}_i) = 1$ if $y_i = 1$ and $N_{\mathcal{W}_1}(\boldsymbol{x}_i) = 0$ otherwise.
2. $N_{\mathcal{W}_2}(\boldsymbol{x}_i) = 1$ if $y_i = -1$ and $N_{\mathcal{W}_2}(\boldsymbol{x}_i) = 0$ otherwise.

Then $(N_{\mathcal{W}_1} - N_{\mathcal{W}_2})(\boldsymbol{x}_i) = y_i$ and $N_{\mathcal{W}_1} - N_{\mathcal{W}_2} = \mathcal{N}_{\tilde{\mathcal{W}}}$ for $\tilde{\mathcal{W}} = (\tilde{W}, \tilde{v})$ where $\tilde{W} \in \mathbb{R}^{2k \times d}$ and $\tilde{\boldsymbol{v}} = (\overbrace{\tilde{v} \ldots \tilde{v}}^{k}, \overbrace{-\tilde{v} \cdots - \tilde{v}}^{k}) \in \mathbb{R}^{2k}, \tilde{v} > 0$.

### 9.3 MISSING PROOFS FOR SECTION 7

#### 9.3.1 PROOF OF THEOREM 8

We first need the following lemma.

**Lemma 11.** *There exists $\hat{\boldsymbol{w}} \in \mathbb{R}^d$ that satisfies the following:*

1. *There exists $\alpha > 0$ such that for each $(\boldsymbol{x}, y) \in S$ we have $|\langle \boldsymbol{x}, \hat{\boldsymbol{w}} \rangle| > \alpha$.*

2. $\#\{(\boldsymbol{x}, y) \in S : \langle \hat{\boldsymbol{w}}, \boldsymbol{x} \rangle < 0\} > \frac{1}{2}|S|.$

*Proof.* Consider the set $V = \{\boldsymbol{v} \in \mathbb{R}^d : \exists_{(\boldsymbol{x},y) \in S} \langle \boldsymbol{v}, \boldsymbol{x} \rangle = 0\}$. Clearly, $V$ is a finite union of hyperplanes and therefore has measure zero, so there exists $\hat{\boldsymbol{w}} \in \mathbb{R}^d \setminus V$. Let $\beta = \min_{(x,y) \in S}\{|\langle \hat{\boldsymbol{w}}, \boldsymbol{x} \rangle|\}$, and since $S$ is finite we clearly have $\alpha > 0$. Finally, if

$$\#\{(\boldsymbol{x}, y) \in S : \langle \hat{\boldsymbol{w}}, \boldsymbol{x} \rangle < 0\} > \frac{1}{2}|S|$$

we can choose $\hat{\boldsymbol{w}}$ and $\alpha = \frac{\beta}{2}$ and we are done. Otherwise, choosing $-\hat{\boldsymbol{w}}$ and $\alpha = \frac{\beta}{2}$ satisfies all the assumptions of the lemma. $\square$

We are now ready to prove the theorem. Choose $\hat{\boldsymbol{w}} \in \mathbb{R}^d$ that satisfies the assumptions in Lemma 11. Now, let $c > \frac{\|\boldsymbol{w}^*\|}{\alpha}$, and let $\boldsymbol{w} = c\hat{\boldsymbol{w}} + \boldsymbol{w}^*$ and $\boldsymbol{u} = c\hat{\boldsymbol{w}} - \boldsymbol{w}^*$. Define

$$W = [\overbrace{\boldsymbol{w} \ldots \boldsymbol{w}}^{k}, \overbrace{\boldsymbol{u} \ldots \boldsymbol{u}}^{k}]^{\top} \in \mathbb{R}^{2k \times d}$$

---

[11] This is where we use the independence assumption on $S$ and $W_0$. In the proof of Theorem 30.2 in Shalev-Shwartz & Ben-David (2014), the hypothesis $h_I$ needs to be independent of $V$. Our independence assumption ensures that this holds.

Let $(\boldsymbol{x}, y) \in S$ be an arbitrary example.

If $\langle \hat{\boldsymbol{w}}, \boldsymbol{x} \rangle > \alpha$, then

$$\langle \boldsymbol{w}, \boldsymbol{x} \rangle = c \langle \hat{\boldsymbol{w}}, \boldsymbol{x} \rangle + \langle \boldsymbol{w}^*, \boldsymbol{x} \rangle \geq c\alpha - \|\boldsymbol{w}^*\| > 0$$
$$\langle \boldsymbol{u}, \boldsymbol{x} \rangle = c \langle \hat{\boldsymbol{w}}, \boldsymbol{x} \rangle - \langle \boldsymbol{w}^*, \boldsymbol{x} \rangle \geq c\alpha - \|\boldsymbol{w}^*\| > 0$$

It follows that

$$\begin{aligned} N_{\mathcal{W}}(\boldsymbol{x}) &= \sum_{1}^{k} \sigma(\langle \boldsymbol{w}, \boldsymbol{x} \rangle) - \sum_{1}^{k} \sigma(\langle \boldsymbol{u}, \boldsymbol{x} \rangle) \\ &= \sum_{1}^{k} (c \langle \hat{\boldsymbol{w}}, \boldsymbol{x} \rangle + \langle \boldsymbol{w}^*, \boldsymbol{x} \rangle) - \sum_{1}^{k} (c \langle \hat{\boldsymbol{w}}, \boldsymbol{x} \rangle - \langle \boldsymbol{w}^*, \boldsymbol{x} \rangle) \\ &= 2k \langle \boldsymbol{w}^*, \boldsymbol{x} \rangle \end{aligned}$$

Therefore $y N_{\mathcal{W}}(\boldsymbol{x}) > 1$, so we get zero loss for this example, and therefore the gradient of the loss will also be zero.

If, on the other hand, $\langle \hat{\boldsymbol{w}}, \boldsymbol{x} \rangle < -\alpha$, then

$$\langle \boldsymbol{w}, \boldsymbol{x} \rangle = c \langle \hat{\boldsymbol{w}}, \boldsymbol{x} \rangle + \langle \boldsymbol{w}^*, \boldsymbol{x} \rangle \leq -c\alpha + \|\boldsymbol{w}^*\| < 0$$
$$\langle \boldsymbol{u}, \boldsymbol{x} \rangle = c \langle \hat{\boldsymbol{w}}, \boldsymbol{x} \rangle - \langle \boldsymbol{w}^*, \boldsymbol{x} \rangle \leq -c\alpha + \|\boldsymbol{w}^*\| < 0$$

and therefore

$$N_{\mathcal{W}}(\boldsymbol{x}) = \sum_{1}^{k} \sigma(\langle \boldsymbol{w}, \boldsymbol{x} \rangle) - \sum_{1}^{k} \sigma(\langle \boldsymbol{u}, \boldsymbol{x} \rangle) = 0.$$

In this case the loss on the example would be $\max\{1 - y N_{\mathcal{W}}(\boldsymbol{x}), 0\} = 1$, but the gradient will also be zero. Along with assumption 2, we would conclude that:

$$L_S(W) > \frac{1}{2}, \ \frac{\partial}{\partial W} L_S(W) = 0$$

Notice that since all the inequalities are strong, the following holds for all $W' \in \mathbb{R}^{2k \times d}$ that satisfies $\|W' - W\| < \epsilon$, for a small enough $\epsilon > 0$. Therefore, $W \in \mathbb{R}^{2k \times d}$ is indeed a local minimum.

### 9.3.2 Proof of Theorem 9

Denote $W_t = [\boldsymbol{w}_t^{(1)} \ldots \boldsymbol{w}_t^{(k)} \boldsymbol{u}_t^{(1)} \ldots \boldsymbol{u}_t^{(k)}]$ and define $K_t = \{\boldsymbol{e}_j : \ \forall_{i \in [k]} \left\langle \boldsymbol{w}_t^{(i)}, \boldsymbol{e}_j \right\rangle \leq 0\}$. We first prove the following lemma.

**Lemma 12.** *For every $t$ we get $K_{t+1} = K_t$.*

*Proof.* Let $\boldsymbol{e}_j$ be the example seen in time $t$. If $N_{\mathcal{W}_t}(\boldsymbol{e}_j) \geq 1$ then there is no update and we are done. Otherwise, if $\boldsymbol{e}_j \in K_t$ then for each $i \in [k]$ we have $\frac{\partial}{\partial \boldsymbol{w}_t^{(i)}} N_{\mathcal{W}_t}(\boldsymbol{e}_j) = 0$ and therefore the update does not change the value of $\boldsymbol{w}_t^{(i)}$, and thus $K_{t+1} = K_t$. If $\boldsymbol{e}_j \notin K_t$ then there exists $i \in [k]$ such that $\left\langle \boldsymbol{w}_t^{(i)}, \boldsymbol{e}_j \right\rangle > 0$. In that case, we update $\boldsymbol{w}_{t+1}^{(i)} \leftarrow \boldsymbol{w}_t^{(i)} + \eta \boldsymbol{e}_j$. Now, note that

$$\left\langle \boldsymbol{w}_{t+1}^{(i)}, \boldsymbol{e}_j \right\rangle = \left\langle \boldsymbol{w}_t^{(i)}, \boldsymbol{e}_j \right\rangle + \eta \langle \boldsymbol{e}_j, \boldsymbol{e}_j \rangle > \left\langle \boldsymbol{w}_t^{(i)}, \boldsymbol{e}_j \right\rangle > 0$$

and therefore $\boldsymbol{e}_j \notin K_{t+1}$. Furthermore, for each $\boldsymbol{e}_\ell$ where $\ell \neq j$, by the orthogonality of the vectors we know that for each $i \in [k]$ it holds that

$$\left\langle \boldsymbol{w}_{t+1}^{(i)}, \boldsymbol{e}_\ell \right\rangle = \left\langle \boldsymbol{w}_t^{(i)}, \boldsymbol{e}_\ell \right\rangle + \eta \langle \boldsymbol{e}_j, \boldsymbol{e}_\ell \rangle = \left\langle \boldsymbol{w}_t^{(i)}, \boldsymbol{e}_\ell \right\rangle$$

Thus $\boldsymbol{e}_\ell \in K_t$ if and only if $\boldsymbol{e}_\ell \in K_{t+1}$ and this concludes the lemma. □

We can now prove the theorem. For each $j \in [d]$, by the symmetry of the initialization, with probability $\frac{1}{2}$ over the initialization of $\boldsymbol{w}_0^{(i)}$, we get that $\left\langle \boldsymbol{w}_0^{(i)}, \boldsymbol{e}_j \right\rangle \leq 0$. Since all $\boldsymbol{w}_i$'s are initialized independently, we get that:

$$P(\boldsymbol{e}_j \in K_0) = P(\cap_{i \in [k]} \left\langle \boldsymbol{w}_0^{(i)}, \boldsymbol{e}_j \right\rangle \leq 0) = \prod_{i \in [k]} P(\left\langle \boldsymbol{w}_0^{(i)}, \boldsymbol{e}_j \right\rangle \leq 0) = \frac{1}{2^k}$$

Now, assuming $k \leq \log_2(\frac{d}{-\ln(\delta)})$, from the independence of the initialization of $\boldsymbol{w}_0^{(i)}$'s coordinates we get

$$P(\cap_{j \in [d]} \boldsymbol{e}_j \notin K_0) = \prod_{j \in [d]} P(\boldsymbol{e}_j \notin K_0)$$

$$= (1 - \frac{1}{2^k})^d \leq e^{-\frac{d}{2^k}} \leq \delta$$

Therefore, with probability at least $1 - \delta$, there exists $j \in [k]$ for which $\boldsymbol{e}_j \in K_0$. By Lemma 12, this implies that for all $t \in \mathbb{N}$ we will get $\boldsymbol{e}_j \in K_t$, and therefore $N_{\mathcal{W}_t}(\boldsymbol{e}_j) \leq 0$. Since $\boldsymbol{e}_j$ is labeled 1, this implies that $L_S(W) > 0$. By the separability of the data, and by the convergence of the SGD algorithm, this implies that the algorithm converges to a stationary point that is not a global minimum. Note that convergence to a saddle point is possible only if we define $\sigma'(0) = 0$, and for all $i \in [k]$ we have at the time of convergence $\left\langle \boldsymbol{w}_t^{(i)}, \boldsymbol{e}_j \right\rangle = 0$. This can only happen if $\left\langle \boldsymbol{w}_0^{(i)}, \boldsymbol{e}_j \right\rangle = \eta N$ for some $N \in \mathbb{N}$, which has probability zero over the initialization of $\boldsymbol{w}_t^{(i)}$. Therefore, the convergence is almost surely to a non-global minimum point.

On the other hand, assuming $k \geq \log_2(\frac{d}{\delta})$, using the union bound we get:

$$P(\cup_{j \in [d]} \boldsymbol{e}_j \in K_0) \leq \sum_{j \in [d]} P(\boldsymbol{e}_j \in K_0)$$

$$= \frac{d}{2^k} \leq \delta$$

So with probability at least $1 - \delta$, we get $K_0 = \emptyset$ and by Lemma 12 this means $K_t = \emptyset$ for all $t \in \mathbb{N}$. Now, if $\boldsymbol{e}_j \notin K_t$ for all $t \in \mathbb{N}$, then there exists $i \in [k]$ such that $\left\langle \boldsymbol{w}_t^{(i)}, \boldsymbol{e}_j \right\rangle > 0$ for all $t \in \mathbb{N}$. If after performing $T$ update iterations we have updated $N > \max\{\frac{C}{\eta}, \frac{1}{\eta}\}$ times on $\boldsymbol{e}_j$, then clearly:

$$\left\langle \boldsymbol{w}_t^{(i)}, \boldsymbol{e}_j \right\rangle = \left\langle \boldsymbol{w}_0^{(i)}, \boldsymbol{e}_j \right\rangle + \sum_{t=0}^{T} \eta \left\langle \boldsymbol{e}_j, \boldsymbol{e}_j \right\rangle \geq \left\langle \boldsymbol{w}_0^{(i)}, \boldsymbol{e}_j \right\rangle + N\eta > 1$$

$$\forall_{i \in [k]} \, s.t \, \left\langle \boldsymbol{u}_0^{(i)}, \boldsymbol{e}_j \right\rangle > 0, \, \left\langle \boldsymbol{u}_t^{(i)}, \boldsymbol{e}_j \right\rangle = \left\langle \boldsymbol{u}_0^{(i)}, \boldsymbol{e}_j \right\rangle - \sum_{t=0}^{T} \eta \left\langle \boldsymbol{e}_j, \boldsymbol{e}_j \right\rangle \leq C - N\eta \leq 0$$

and therefore $N_{\mathcal{W}_t}(\boldsymbol{e}_j) > 1$, which implies that $L_{\{(\boldsymbol{e}_j, 1)\}}(W_t) = 0$. From this, we can conclude that for each $j \in [d]$, we perform at most $\lceil \max\{\frac{C}{\eta}, \frac{1}{\eta}\} \rceil$ update iterations on $\boldsymbol{e}_j$ before reaching zero loss, and therefore we can perform at most $\lceil \max\{\frac{dC}{\eta}, \frac{d}{\eta}\} \rceil$ update iterations until convergence. Since we show that we never get stuck with zero gradient on an example with loss greater than zero, this means we converge to a global optimum after at most $\lceil \max\{\frac{dC}{\eta}, \frac{d}{\eta}\} \rceil$ iterations.

