# OpenReview forum: "SGD Learns Over-parameterized Networks that Provably Generalize on Linearly Separable Data"
_ICLR.cc/2018/Conference — Accept (Poster)_

### Official Review · AnonReviewer2 · 2017-11-25
**Interesting result for generalisation guarantees of overparametrised 1-hidden layer network with fixed output layer.**

**Rating:** 7
**Confidence:** 3

**Review:**

Paper studies an interesting phenomenon of overparameterised models being able to learn well-generalising solutions. It focuses on a setting with three crucial simplifications:
- data is linearly separable
- model is 1-hidden layer feed forward network with homogenous activations
- **only input-hidden layer weights** are trained, while the hidden-output layer's weights are fixed to be (v, v, v, ..., v, -v, -v, -v, ..., -v) (in particular -- (1,1,...,1,-1,-1,...,-1))
While the last assumption does not limit the expressiveness of the model in any way, as homogenous activations have the property of f(ax)=af(x) (for positive a) and so for any unconstrained model in the second layer, we can "propagate" its weights back into first layer and obtain functionally equivalent network. However, learning dynamics of a model of form
 z(x) = SUM( g(Wx+b) ) - SUM( g(Vx+c) ) + d
and "standard" neural model
 z(x) = Vg(Wx+b)+c
can be completely different.
Consequently, while the results are very interesting, claiming their applicability to the deep models is (at this point) far fetched. In particular, abstract suggests no simplifications are being made, which does not correspond to actual result in the paper. The results themselves are interesting, but due to the above restriction it is not clear whether it sheds any light on neural nets, or simply described a behaviour of very specific, non-standard shallow model.

I am happy to revisit my current rating given authors rephrase the paper so that the simplifications being made are clear both in abstract and in the text, and that (at least empirically) it does not affect learning in practice. In other words - all the experiments in the paper follow the assumption made, if authors claim is that the restriction introduced does not matter, but make proofs too technical - at least experimental section should show this. If the claims do not hold empirically without the assumptions made, then the assumptions are not realistic and cannot be used for explaining the behaviour of models we are interested in.

Pros:
- tackling a hard problem of overparametrised models, without introducing common unrealistic assumptions of activations independence
- very nice result of "phase change" dependend on the size of hidden layer in section 7

Cons:
- simplification with non-trainable second layer is currently not well studied in the paper; and while not affecting expressive power - it is something that can change learning dynamics completely

# After the update

Authors addressed my concerns by:
- making simplification assumption clearer in the text
- adding empirical evaluation without the assumption
- weakening the assumptions

I find these modifications satisfactory and rating has been updated accordingly.

---

### Official Review · AnonReviewer1 · 2017-11-26
**The paper proves interesting properties of SGD on linearly separable data, first result on a interesting direction although the assumption/techniques seem a bit limited.**

**Rating:** 7
**Confidence:** 3

**Review:**

This paper shows that on linearly separable data, SGD on a overparametrized network (one hidden layer, with leaky ReLU activations) can still lean a classifier that provably generalizes. The assumption on data and structure of network is a bit strong, but this is the first result that achieves a number of desirable properties
``1. Works for overparametrized network
2. Finds global optimal solution for a non-convex network.
3. Has generalization guarantees (and generalization is related to the SGD algorithm).
4. Number of samples need not depend on the number of neurons.

There have been several papers achieving 1 and 2 (with much weaker assumptions), but they do not have 3 and 4. The proof of the optimization part is very similar to the proof of perceptron algorithm, and really relies on linear separability. The proof of generalization is based on a compression argument, where if an algorithm does not take many nonzero steps, then it must have good generalization. Ideally, one would also want to see a result where overparametrization actually helps (in the main result the whole data can be learned by a linear classifier). This is somewhat achieved when the activation is replaced with standard ReLU, where the paper showed with a small number of hidden units the algorithm is likely to get stuck at a local minima, but with enough hidden units the algorithm is likely to converge (but even in this case, the data is still linearly separable and can be learned just by a perceptron).

The main concern about the paper is the possibility of generalizing the result. The algorithm part seems to heavily rely on the linear separable assumption. The generalization part relies on not making many non-zero updates, which is not really true in realistic settings (where the data is accessed in multiple passes) [After author response: Yes in the linearly separable case with hinge loss it is quite possible that the number of updates is sublinear. However what I meant here is that with more complicated data and different loss functions it is hard to believe that this can still hold.]. The related work section is also a bit unfair to some of the other generalization results (e.g. Bartlett et al. Neyshabur et al.): those results work on more general network settings, and it's not completely clear that they cannot be related to the algorithm because they rely on certain solution specific quantities (such as spectral/Frobenius norms of the weight matrices) and it could be possible that SGD tends to find a solution with small norm (which can be proved in linear setting and might also be provable for the setting of this paper) [This is addressed in the author response].

Overall, even though the assumptions might be a bit strong, I think this is an interesting result working towards a good direction and should be accepted.

---

### Official Review · AnonReviewer3 · 2017-12-01
**Good paper on understanding the role of SGD in generalization**

**Rating:** 8
**Confidence:** 4

**Review:**

Summary:
This paper considers the problem of classifying linearly separable data with a two layer \alpha- Leaky ReLU network, in the over-parametrized setting with 2k hidden units. The algorithm used for training is SGD which minimizes the hinge loss error over the training data. The parameters in the top layer are fixed in advance and only the parameters in the hidden layer are updated using SGD. First result shows that the loss function does not have any sub-optimal local minima. Later, for the above method, the paper gives a bound proportional to ||w*||^2/\alpha^2, on the number of non-zero updates made by the algorithm (similar to perceptron analysis), before converging to a global minima - w*. Using this a generalization error bound independent of number of hidden units is presented. Later the paper studies ReLU networks and shows that loss in this case can have sub-optimal local minima.

Comments:

This paper considers a simpler setting to study why SGD is successful in recovering solutions that generalize well even though the neural networks used are typically over-parametrized. While the paper considers a simpler setting of classifying linearly separable data and training only the hidden layer, it nevertheless provides a useful insight on the role of SGD in recovering solutions that generalize well (independent of number of hidden units 'k').

One confusing aspect in the paper is the optimization and generalization results hold for any global minima w* of the L_s(w). There is a step missing of taking the minimum over all such w*, which will give the tightest bounds for SGD, and it will be useful to clear this up in the paper.

More importantly I am curious how close the updates are when, 1)SGD is updating only the hidden units  and 2) SGD is updating both the layers. Simple intuition suggests SGD might update the top layer "more" that the hidden layer as the gradients tend to decay down the layers. It is useful to discuss this in the paper and may be have some experiments on linearly separable data but with updates in both layers.

---

### Author Response · Authors · 2017-12-20
**Response to All Reviewers**

We thank the reviewers for their helpful feedback. The main concern that was raised by the reviewers is whether these results generalize to a realistic neural network training process.
Specifically, in the submission we have analyzed a variant of SGD which updates only the first layer of the network, while keeping the weights of the second layer fixed. AnonReviewer2 correctly notes that in practice, the second layer is also updated, and asks to what degree our results hold in this case. To address this concern, we revise the text as follows:
1. We clearly state our assumptions in both the abstract and the paper itself (see Section 5.3).
2. We conduct the same experiments as in the paper, but with both layers trained. We empirically show that training both layers has similar training and generalization performance as training the first layer (Figure 2).
3. We show that the main theoretical result still holds even when the second layer weights are updated, as long as they do not change signs during the training process, and their absolute values are bounded from below and from above.
4. We conduct experiments similar to the setting in (2) above, but now we choose a constant step size such that the condition in (3) above holds. Namely, we ensure that the weights of the last layer do not change their sign, and are correctly bounded. The performance of SGD in this case is similar to previous experiments and is in line with our theoretical findings.

The above show that although the dynamics of the problem indeed change when updating the second layer, our results and conclusions still hold. A complete theoretical analysis of the two layer case is left for future work.

Regarding the linear separability assumption, this is a realistic setting and this assumption allows us to show for the first time a complete analysis of optimization and generalization for over-parameterized neural networks. We are not aware of any other result of this kind under different realistic assumptions.
As for the proposition that SGD tends to find solutions with small norm in our problem, we are not aware of any existing results that imply that this is indeed the case, though this may be an interesting problem to study in the future. We have rephrased our notes on other generalization results in the related work section, addressing AnonReviewer1’s remark.
AnonReviewer1 mentioned that in practice there should be many non-zero updates since the data is accessed multiple times. However, we note that we considered the hinge loss, which vanishes for points that are classified with a margin. Therefore, it is possible that with multiple passes over the data there are only a few non-zero updates.
Finally, AnonReviewer3 notes that we can optimize our bound with respect to w^*. This is true, as in the vanilla Perceptron, the best w* is the one with the largest margin.

---

### Decision · Program_Chairs · 2018-01-29
**ICLR 2018 Conference Acceptance Decision**

**Decision:**

Accept (Poster)

**Comment:**

This is a high quality paper, clearly written, highly original, and clearly significant. The paper gives a complete analysis of SGD in a two layer network where the second layer does not undergo training and the data are linearly separable.  Experimental results confirm the theoretical suggestion that the second layer can be trained provided the weights don't change sign and remain bounded. The authors address the major concerns of the reviewers (namely, whether these results are indicative given the assumptions). This line of work seems very promising.